# The Evolving Paradigm of Antibody–Drug Conjugates Targeting the ErbB/HER Family of Receptor Tyrosine Kinases

**DOI:** 10.3390/pharmaceutics16070890

**Published:** 2024-07-02

**Authors:** Peyton High, Cara Guernsey, Shraddha Subramanian, Joan Jacob, Kendra S. Carmon

**Affiliations:** 1Center for Translational Cancer Research, The Brown Foundation Institute of Molecular Medicine, The University of Texas Health Science Center at Houston, Houston, TX 77030, USA; peyton.high@uth.tmc.edu (P.H.); cara.guernsey@uth.tmc.edu (C.G.); shraddha.subramanian@uth.tmc.edu (S.S.); joan.jacob@uth.tmc.edu (J.J.); 2Graduate School of Biomedical Sciences, The University of Texas MD Anderson Cancer Center and UTHealth Houston, Houston, TX 77030, USA

**Keywords:** antibody–drug conjugates, EGFR, HER2, HER3, bispecific antibodies, combination therapy, drug resistance, receptor tyrosine kinase, therapeutics

## Abstract

Current therapies targeting the human epidermal growth factor receptor (HER) family, including monoclonal antibodies (mAbs) and tyrosine kinase inhibitors (TKIs), are limited by drug resistance and systemic toxicities. Antibody–drug conjugates (ADCs) are one of the most rapidly expanding classes of anti-cancer therapeutics with 13 presently approved by the FDA. Importantly, ADCs represent a promising therapeutic option with the potential to overcome traditional HER-targeted therapy resistance by delivering highly potent cytotoxins specifically to HER-overexpressing cancer cells and exerting both mAb- and payload-mediated antitumor efficacy. The clinical utility of HER-targeted ADCs is exemplified by the immense success of HER2-targeted ADCs including trastuzumab emtansine and trastuzumab deruxtecan. Still, strategies to improve upon existing HER2-targeted ADCs as well as the development of ADCs against other HER family members, particularly EGFR and HER3, are of great interest. To date, no HER4-targeting ADCs have been reported. In this review, we extensively detail clinical-stage EGFR-, HER2-, and HER3-targeting monospecific ADCs as well as novel clinical and pre-clinical bispecific ADCs (bsADCs) directed against this receptor family. We close by discussing nascent trends in the development of HER-targeting ADCs, including novel ADC payloads and HER ligand-targeted ADCs.

## 1. Introduction

The human epidermal growth factor receptor (HER) family is an important class of receptor tyrosine kinases (RTKs) consisting of the epidermal growth factor receptor (ErbB1/EGFR), ErbB2/HER2, ErbB3/HER3, and ErbB4/HER4. The HER family signals through a variety of substrates with integral roles in normal cellular processes including proliferation, differentiation, development, migration, adhesion, and others [1]. HER family members are frequently hyperactivated through mutation or overexpression in human cancers [2]. In the context of cancer, the HER family promotes uncontrolled cell growth, tumor progression, and metastasis. Consequently, there have been extensive efforts to target this family therapeutically, with the most successful avenues being monoclonal antibodies (mAbs) and tyrosine kinase inhibitors (TKIs) against EGFR and HER2. Yet, intrinsic and acquired drug resistance are frequently observed in response to these therapies, limiting their efficacy to particular patient subsets and emphasizing the need for novel therapeutic approaches [3,4]. Recently, antibody–drug conjugates (ADCs) have emerged as a rapidly growing class of anti-cancer therapeutics to target HER family members and other RTKs [5]. Overall, 2 of 13 FDA-approved ADCs target HER2 and more than 50 of the 100 plus ADCs in clinical trials are against RTKs, including EGFR, HER2, HER3, c-MET, AXL, ROR1/2, EPHA3/5, FGFR3, and IGF-1R [6]. In this review, we provide a comprehensive evaluation of mono- and bispecific ADCs targeting each of the HER family members and emergent trends in HER-targeting ADCs, including novel payloads and HER ligand-targeting ADCs.

### 1.1. HER Activation and Signaling Mechanisms

Prior to ligand binding, EGFR, HER3, and HER4 monomers exist on the cell surface in an autoinhibitory state in which their dimerization arms are inaccessible to other cognate receptors [7,8]. The dimerization arm of HER2, rather, is constitutively exposed, deeming HER2 the preferred dimerization partner for all other HER monomers [9]. Notably, HER2 does not form homodimers in response to ligands but is able to form homodimers under states of HER2 overexpression [10]. Ligand binding to EGFR, HER3, or HER4 promotes a conformational change that exposes the receptor dimerization arm to interact with HER2 or a second ligand-bound receptor. Additionally, an increasing number of reports have described the existence of pre-formed, inactive EGFR dimers that, upon ligand binding, undergo a conformational change to reorient and activate the intracellular kinase domain [11]. There are seven HER family ligands including heparin-binding EGF (HB-EGF), amphiregulin (AREG), epiregulin (EREG), epigen (EPGN), neuregulin (NRG1-4), betacellulin (BTC), and transforming growth factor alpha (TGFɑ). EGF, TGFɑ, AREG, and EPGN exclusively bind EGFR; HB-EGF, EREG, and BTC are able to bind EGFR or HER4; NRG1/2 can bind HER3 and HER4; and NRG3/4 bind HER4 [12]. HER2 has no known natural ligand [13]. Following ligand binding, the activating kinase domain of one monomer phosphorylates the receiving kinase domain of the other monomer, termed trans-autophosphorylation [14]. Notably, while HER3 heterodimers are functionally active, HER3 homodimers possess minimal autophosphorylation because of mutations in its kinase domain [15]. Phosphorylated tyrosine residues in the C-terminal tail serve as docking sites for Src homology 2 (SH2) or phosphotyrosine-binding (PTB) domain-containing adaptor proteins that potentiate RAS/MAPK, PI3K/AKT, and PLCγ/PKC pathways, among others [16]. Importantly, different heterodimers, ligands, and phosphotyrosine residues confer bias toward different signaling pathways with varying levels of activation [17,18].

### 1.2. The HER Family in Cancer

HER family signaling is frequently dysregulated in cancers. For example, EGFR point mutations such as L858R and the T790M “gatekeeper” mutation, as well as exon 19 in-frame deletions, are all commonly observed in lung adenocarcinomas [19,20]. Additionally, deletion of exons 2–7 in the EGFR extracellular domain (ECD) results in a constitutively active “EGFRvIII” mutant frequently observed in glioblastoma multiforme (GBM) [21]. EGFR amplification is also broadly observed across cancer types, including 60–80% of colorectal cancers (CRCs) [22,23,24,25]. HER2 overexpression, found in 15–25% of breast cancers (BCs) and 10–30% of gastric cancers (GCs), is a common mechanism to promote cancer progression via the formation of ligand-independent, constitutively active HER2 homodimers [26,27]. HER3 is overexpressed in 50–70% of BCs, where HER2/HER3 heterodimers serve as functional tumor drivers [15,28,29]. The roles of HER4 in cancer are less well-studied, with both oncogenic and tumor-suppressive functions having been documented [30,31,32,33,34,35,36]. Taken together, the implications of the HER family in mediating cancer progression have prompted the development of several therapeutics against this family.

### 1.3. HER-Targeted TKIs and mAbs: Successes and Challenges

TKIs were the first therapeutic agents designed to target the HER family. TKIs compete for the ATP binding site in the receptor catalytic domain, inhibiting receptor activation and downstream signaling [37]. The first EGFR-targeting TKI, gefitinib, was approved in 2003 for non-small cell lung cancer (NSCLC), followed by erlotinib for NSCLC and pancreatic cancer in 2004 and 2005, respectively [38,39,40]. Notably, despite both exhibiting reversible binding to the EGFR tyrosine kinase domain, erlotinib is able to bind to both active and inactive EGFR conformations, whereas gefitinib only binds the active EGFR conformation [41]. While these first-generation EGFR TKIs are initially efficacious, nearly all patients develop resistance due to acquired mutations in EGFR (i.e., T790M), downstream mutations in KRAS/BRAF/PIK3CA, and alternative RTK activation, such as mesenchymal–epithelial transition factor (c-MET) [4,42]. Because of resistance mutations that allow evasion of first-generation TKIs, osimertinib, which irreversibly binds to residue C797 in the EGFR ATP binding pocket and demonstrates efficacy against T790M-harboring tumors, was approved in 2015 [43,44]. Yet, mechanisms of osimertinib resistance, such as EGFR C797S mutation, are still observed [45]. Other FDA-approved HER-targeting TKIs, including afatinib (EGFR/HER2/HER4), dacomitinib (pan-HER), lapatinib (EGFR/HER2), and neratinib (EGFR/HER2), are subject to similar resistance mechanisms [46,47,48,49].

MAbs are another class of therapeutics that bind HER monomer ECDs [50,51]. HER-targeting mAbs may exert cytostatic or cytotoxic effects through multiple mechanisms that include inhibiting ligand binding, receptor dimerization, and downstream signaling or promoting receptor internalization and degradation. MAbs may also engage Fcγ receptors on immune cells, resulting in antibody-dependent cellular cytotoxicity (ADCC) and antibody-dependent cellular phagocytosis (ADCP) [52]. There are currently six FDA-approved mAbs targeting EGFR or HER2 and ongoing phase I/II clinical trials for HER3-targeting mAbs (NCT05057013, NCT04383210, NCT05203601, NCT05910827) [53,54,55,56]. EGFR-targeting mAbs are approved for various oncologic indications and all share similar binding epitopes in the EGFR ECDIII as follows: cetuximab (mouse/human chimeric mAb approved for metastatic CRC and head and neck squamous cell carcinoma (HNSCC)), panitumumab (human mAb approved for metastatic CRC), and necitumumab (human mAb approved for NSCLC). HER2-targeting mAbs are also approved for various oncologic indications, though they target slightly different epitopes as follows: trastuzumab (humanized mAb approved for BC, GC, and gastroesophageal junction adenocarcinoma; binds HER2 ECDII), pertuzumab (humanized mAb approved for BC; binds HER2 ECDII separate from trastuzumab), and margetuximab (mouse/human chimeric Fc-modified mAb approved for BC; binds same HER2 ECDII as trastuzumab) [57,58]. Yet, mAbs are still subject to the aforementioned mechanisms of resistance as TKIs as well as impaired ADCC/ADCP and dysregulated receptor degradation [59,60]. Example resistance mechanisms are depicted in Figure 1. Bispecific antibodies (bsAbs) that target combinations of EGFR, HER2, and HER3 or other surface receptors such as hepatocyte growth factor receptor (c-MET) and programmed cell death protein 1 (PD-1) have been developed to circumvent some of these resistance mechanisms, though resistance to these modalities is also not uncommon. For example, analysis of patient responses to amivantamab, an EGFR/c-MET bsAb approved for NSCLC, revealed resistance mechanisms including c-MET/EGFR amplification and *PIK3CA* mutations [61]. Considering the pervasive resistance mechanisms associated with current HER-targeting therapies, there is still a need for alternative strategies to target this family.

### 1.4. Antibody–Drug Conjugates: The Anti-Cancer Biological Missiles

ADCs are a rapidly expanding class of anti-cancer therapeutics with 13 currently FDA-approved, including six for solid tumors and seven for hematological malignancies, and over 100 more in clinical trials [6]. ADCs consist of a mAb backbone conjugated to potent cytotoxins via a cleavable or non-cleavable linker. Cleavable linkers are more common among approved and preclinical ADCs and are cleaved by a variety of different mechanisms including lysosomal protease activity (e.g., cathepsin B), tumor-associated acidic/reducing conditions, or other enzymatic cleavage (i.e., beta-glucuronidase, sulfatases, phosphatases) [62]. ADCs with non-cleavable linkers, rather, require lysosomal degradation of the entire ADC for payload release. ADCs target surface-expressed, tumor-enriched antigens, enabling specific delivery of payloads to cancer cells and sparing normal tissue. The therapeutic index (TI) of ADC payloads is, in theory, enhanced compared with traditional chemotherapies by restricting their action to cancer cells. Among approved ADCs, traditional payload classes include DNA-damaging agents (i.e., calicheamicins and pyrrolobenzodiazepines (PBDs)); microtubule-disrupting auristatins and maytansanoids (i.e., monomethyl auristatin E (MMAE), monomethyl auristatin F (MMAF), DM1, and DM4); and topoisomerase I inhibitors (i.e., SN38 and DXd) [63]. While topoisomerase I inhibitors are the most predominant payload class in clinical trials, there is an increasing amount of alternative payloads being examined including topoisomerase II and RNA polymerase II inhibitors, immune agonists, proteolysis-targeting activating chimeras (PROTACs), and TKIs [64]. ADC payload is the most significant determinant of clinical toxicity profiles because of non-specific, normal tissue uptake [65]. Common treatment-related adverse events (TRAEs) include peripheral neuropathy and hematological malignancies for MMAE ADCs, ocular toxicity for DM4 and MMAF ADCs, thrombocytopenia and liver toxicity for DM1 ADCs, and hematological and gastrointestinal malignancies for PBD ADCs [66]. In response, several dose optimization and management strategies such as capping of treatment duration, fractionated dosing, treatment response-guided dose adjustments, and randomized dose-finding studies have all been employed [65]. Furthermore, innovations in ADC conjugation techniques have been made to improve TI and tolerability. Compared with traditional stochastic cysteine- or lysine-based conjugation strategies that result in heterogeneous pools of ADCs with varying DARs, site-specific and tag-free enzymatic conjugation methods have recently emerged as a promising strategy to generate homogenous ADC pools with improved pharmacodynamic and pharmacokinetic properties [67,68]. Upon binding its target antigen, ADCs are internalized and trafficked to the lysosome, resulting in linker cleavage, drug release, and antigen degradation [69]. Additionally, many ADC payloads are non-polar and diffuse into neighboring tumor cells to exert target-independent cytotoxicity, termed the “bystander effect,” though there are notable exceptions to this (MMAF and DM1) [70]. Furthermore, ADCs retain the therapeutic effects of the parent mAb. Therefore, the anti-tumor mechanisms of RTK-targeting ADCs may include (1) neutralization of downstream signaling; (2) Fcγ receptor-mediated immune cell engagement; (3) payload-mediated cytotoxicity; and (4) antigen degradation (Figure 2). Importantly, since ADC payloads exert cytotoxic effects independent of mAb and TKI cell-killing mechanisms, ADCs may be an effective option to overcome traditional resistance mechanisms of HER-targeted therapies.

A wide variety of novel ADC targets are being examined in clinical trials including RTKs, G protein-coupled receptors, glycoproteins, adhesion receptors, and immune checkpoint proteins [71,72,73,74,75,76]. The rapid growth of HER2-targeting ADCs exemplifies the potential for clinical success of HER-targeting ADCs. HER2 is the most popular ADC target, with 2 HER2-targeting ADCs having received FDA approval and 26 more in the clinical pipeline [6]. The two next most targeted RTKs lag far behind HER2, with seven c-MET- and four EGFR-targeted ADCs currently in clinical trials. The remainder of this review focuses on selected approved and clinically relevant HER-targeting monospecific and bispecific ADCs (bsADCs), as summarized in Table 1 and Table 2, respectively. Notably, no HER4-targeting ADCs have been reported to date.

## 2. Monospecific ADCs Targeting the ErbB Receptor Family

### 2.1. EGFR-Targeting ADCs

As previously discussed, EGFR amplification and oncogenic mutations are commonly implicated in tumorigenesis, disease progression, and drug resistance. While there are several FDA-approved EGFR-targeting mAbs and TKIs, their efficacy is limited. EGFR-targeting ADCs have yet to receive FDA approval, though there are several currently under clinical investigation (Table 1) and in preclinical development.

#### 2.1.1. Depatuxizumab Mafodotin (ABT-414)

Depatuxizumab mafodotin, or ABT-414, developed by Abbvie, consists of an ABT-806 mAb backbone conjugated via interchain cysteines to the microtubule-inhibiting payload MMAF via a non-cleavable maleimidocaproyl (mc) linker with an average drug-to-antibody ratio (DAR) of 3.8 [77,78]. Importantly, ABT-806 exhibits nearly 10-fold greater affinity for overexpressed, tumor-associated EGFR versus WT-EGFR. Phase I clinical trials (NCT01472003) of the ABT-806 mAb showed it was well-tolerated with significant brain uptake and minimal cutaneous toxicity, deeming it a rational choice for ADC development for brain tumors [79]. Preclinically, ABT-414 promoted tumor growth inhibition (TGI) and regression superior to ABT-806 in several EGFR-overexpressing xenografts [78]. In particular, ABT-414 was highly effective in GBM xenografts (2–10 mg/kg) and strongly synergized with the DNA-alkylating agent temozolomide (TMZ) and fractionated radiation at a low ADC dose (1 mg/kg). ABT-414 was assessed in phase I trials for EGFR-amplified, recurrent GBM (NCT01800695), wherein the most commonly observed TRAEs were reversible and ocular (i.e., blurred vision and dry eye), as is common with the MMAF payload [80,168]. Importantly, ABT-414 demonstrated reasonable efficacy with a median progression-free survival (PFS) of 2.1 months. ABT-414 was further evaluated in phase II trials at 1.25 mg/kg in adult participants with recurrent GBM with or without TMZ versus TMZ/lomustine monotherapies (NCT02343406). Combination ABT-414 and TMZ treatment compared with TMZ/lomustine monotherapy increased median PFS and overall survival (OS), warranting the investigation of ABT-414 in combination with standard chemo-irradiation and TMZ in phase III trials for newly diagnosed EGFR-amplified GBM patients (NCT02573324).

Unfortunately, ABT-414 phase III trials were discontinued for futility purposes with one potential explanation being limited tumor and blood–brain barrier (BBB) penetration [81]. Regarding tumor penetration, preclinical and clinical data showed that ABT-414 efficacy and intratumoral concentration were inversely correlated with tumor size (NCT01800695) [82]. Furthermore, while the ABT-806 mAb showed significant brain uptake in clinical studies, post-phase III trial ABT-414 studies demonstrated that GBM patient-derived xenografts (PDXs) intrinsically sensitive to ABT-414 as subcutaneous tumors were significantly less responsive when engrafted as orthotopic, intracranial tumors (2/7 responders). Furthermore, sensitivity was directly correlated with ABT-414 brain tumor uptake, suggesting ABT-414 BBB penetration may hinder efficacy [83]. Consistently, ABT-414 direct intratumoral injection restored PDX sensitivity. While the study did not examine ABT-806 uptake alongside ABT-414, it would be intriguing to evaluate how the addition of the linker–payload moiety affects biodistribution, BBB penetration, and, therefore, ADC efficacy. Beyond strategies to enhance BBB/tumor penetration, higher affinity mAbs with bystander effect–competent payloads may represent more effective approaches to GBM-indicated ADCs. Accordingly, additional ADCs were developed by Abbvie utilizing an affinity-matured ABT-806 (AM1) backbone as follows: losatuxizumab vedotin, or ABBV-221 (MMAE payload), and serclutamab talirine, or ABBV-321 (PBD payload) [84,85,87]. Unfortunately, phase I trials of ABBV-221 were halted because of the high incidence of infusion reactions, and ABBV-321 phase I trials were closed because of a lack of efficacy (NCT02365662, NCT03234712) [86,87].

#### 2.1.2. MRG003

MRG003, developed by Miracogen, is an EGFR-targeting ADC consisting of an MMAE payload and a cleavable valine–citrulline (vc) linker. Phase I trials for MRG003 were completed for patients with EGFR-positive tumors (NCT04868344), and reasonable objective response rates (ORRs) were observed for HNSCC (40%) and nasopharyngeal carcinoma (NPC; 44%), although not for CRC patients (0%) [88]. Inefficacy in CRCs may be due to the relative insensitivity of these tumors to microtubule-destabilizing agents [169]. Thirty-one percent of patients reported ≥ grade 3 treatment-related AEs (TRAEs), including hyponatremia, leukocytopenia, and neutropenia. As observed with ABT-414, MRG003 exhibited less skin toxicity than EGFR mAbs cetuximab or panitumumab. Several phase II trials examining the efficacy of MRG003 monotherapy are underway for solid cancers (NCT04868162, NCT04838548, NCT05188209, NCT04838964, NCT05126719) as well as in combination with pucotenlimab or HX-008, a PD-1-targeting mAb (NCT05688605).

#### 2.1.3. HLX42

HLX42, developed by Henlius, is a next-generation EGFR-targeting ADC with a novel topoisomerase I inhibitor and a DAR of 8 [89]. HLX42 integrates a linker that is cleaved in the tumor microenvironment (TME), circumventing the requirement for ADC internalization. Preclinical studies of HLX42 demonstrated its superior efficacy over an anti-EGFR-GGFG-DXd ADC in lung cancer xenografts (8 mg/kg), supporting the idea that the novel HLX42 linker and topoisomerase payload may confer improved efficacy over validated linker–payloads on FDA-approved ADCs (i.e., GGFG-DXd on trastuzumab deruxtecan). Pilot toxicity studies determined the highest non-severely toxic dose (HNSTD) in nonhuman primates as 20 mg/kg. HLX42 recently began phase I trials for advanced and metastatic solid tumors (NCT06210815).

While an EGFR-targeted ADC has yet to receive FDA approval, candidates like ABT-414 and MRG003 have demonstrated reasonable safety profiles in late-stage clinical trials, a major concern for therapeutic targeting of EGFR due to its ubiquitous expression. Despite ABT-414 inefficacy in phase III trials, MRG003 is the next most-progressed EGFR-targeting ADC and is being evaluated in a variety of EGFR-expressing tumors.

### 2.2. HER2-Targeting ADCs

ADCs targeting HER2 have been rather prolific in the clinical setting, and one of these agents, trastuzumab deruxtecan, recently received tumor-agnostic approval for HER2-positive (IHC3+) tumors [170]. Here, we briefly review recent clinical advances for approved HER2 ADCs and discuss HER2-targeted ADCs in phase III clinical development (Table 1). Additional information on earlier-stage HER2 ADCs may be found elsewhere [171].

#### 2.2.1. Trastuzumab Emtansine (T-DM1)

Trastuzumab emtansine or T-DM1 is a HER2 ADC developed by Genentech in collaboration with ImmunoGen that comprises a trastuzumab backbone conjugated to DM1 via a nonreducible thioether linker (DAR = 3.5) [90,91]. T-DM1 was approved in 2013 for HER2-positive metastatic BC (mBC) based on the EMILIA trial, wherein T-DM1 significantly improved PFS and OS compared with lapatinib plus the antimetabolite capecitabine [92]. T-DM1 was subsequently approved in 2019 as adjuvant therapy for the treatment of HER2-positive early BC in patients with residual invasive disease following taxane (mitotic inhibitor)-based chemotherapy and trastuzumab treatment [93]. This approval was based on results from the KATHERINE trial where three years post-treatment, patients receiving T-DM1 experienced improved rates of invasive disease-free survival (88.3%) versus those on trastuzumab (77%). Common TRAEs associated with T-DM1 include nausea, fatigue, thrombocytopenia, and diarrhea [94]. Multiple trials are ongoing to increase T-DM1 efficacy and expand its utility. The ATEMPT 2.0 phase II trial (NCT04893109), for example, is examining T-DM1 in stage I HER2-positive BC versus the combination of paclitaxel and trastuzumab. Other trials are examining T-DM1 combination with HER2 TKI, tucatinib (NCT03975647); PD-1 mAb, pembrolizumab (NCT03032107); and PD-0332991, a cyclin-dependent kinase 4/6 (CDK4/6) inhibitor (NCT01976169).

#### 2.2.2. Trastuzumab Deruxtecan (T-DXd)

Similar to T-DM1, trastuzumab deruxtecan, or T-DXd, co-developed by Daiichi Sankyo and AstraZeneca, is composed of a trastuzumab backbone. Deruxtecan refers to the linker-drug entity of a cathepsin-cleavable tetrapeptide (GGFG) linker and exatecan derivative DXd payload, which functions via topoisomerase I inhibition (DAR = 8) [95,96]. T-DXd was approved for HER2-positive mBC in 2019 following results from the DESTINY-Breast01 trial, where patients previously treated with T-DM1 achieved a PFS of 16.4 months and an estimated OS of 93.9% after 5.4 mg/kg T-DXd treatment (NCT03248492) [97]. Common ≥ grade 3 TRAEs include decreased neutrophil count, anemia, and nausea, and 13.6% of patients in the DESTINY-Breast01 trial also experienced interstitial lung disease (ILD) thought to be mediated by Fcγ receptor-mediated uptake on alveolar macrophages [98]. DESTINY-Breast03, which directly assessed T-DXd versus T-DM1 in HER2-positive mBC, found that patients receiving T-DXd experienced significantly improved median PFS compared with T-DM1 with similar rates of ≥ grade 3 TRAEs [99]. Furthermore, in patients with brain metastases, T-DXd promoted an intracranial ORR of 65.7% versus 34.3% for T-DM1. Similarly, the DESTINY-Breast05 trial is directly comparing the head-to-head efficacy of T-DXd versus T-DM1 in HER2-positive BC patients with residual disease following neoadjuvant therapy (NCT04622319). Recent clinical approval of T-DXd in HER2-low tumors, a majority subset of advanced BCs without adequate treatments, has significantly altered BC treatment and the ADC paradigm [172]. The DESTINY-Breast04 phase III trial evaluated the efficacy of T-DXd in previously treated, HER2-low (IHC 1+ or ICH2+/ISH−) mBCs (NCT03734029) and showed that T-DXd significantly improved median PFS (9.9 months) versus physician’s choice (5.5 months). T-DXd was therefore approved for unresectable or metastatic, previously treated HER2-low BC. The DESTINY-Breast06 trial aims to further expand this by evaluating hormone receptor-positive (HR+) HER2-low or HER2-ultralow (IHC > 0, <1+) patients (NCT04494425). Importantly, primary results from this trial showed that T-DXd conferred a clinically relevant increase in PFS for both HER2-low and HER2-ultralow patients versus physician’s choice of chemotherapy with no unexpected changes in safety (13.2 months versus 8.1 months and 13.2 months versus 8.3 months for HER2-low and HER2-ultralow, respectively) [100]. This may signal an important paradigm shift in the determinants of ADC efficacy beyond antigen abundance [173].

While T-DM1 is solely approved for BC, the DESTINY-Lung02 (NCT04644237) and DESTINY-Gastric01 (NCT03329690) trials prompted T-DXd approval for HER2-positive NSCLC and GCs. Furthermore, in April 2024, T-DXd also received tumor-agnostic approval in metastatic HER2-positive (IHC3+) solid tumors following results from the DESTINY-PanTumor02, DESTINY-CRC02, and DESTINY-Lung01 trials (NCT04482309; NCT04744831; NCT03505710) [101,102,170]. Like T-DM1, ongoing trials are evaluating the efficacy of T-DXd as a frontline treatment and in combination with other agents. For example, the DESTINY-Breast09 phase III trial (NCT04784715) is evaluating T-DXd as a first-line treatment in HER2-positive BC, either as monotherapy or in combination with pertuzumab. The SATEEN trial is examining the efficacy of sacituzumab govitecan, an FDA-approved TROP2-targeting ADC, in HER2-positive BCs following progression on T-DXd (NCT06100874). Additional trials are evaluating the efficacy of T-DXd with immune therapies and TKIs, such as nivolumab and tucatinib (NCT05480384; NCT02614794).

#### 2.2.3. Disitamab Vedotin (RC48)

Disitamab vedotin, or RC48, developed by Remegen Biosciences and Yantai Rongchang Biological Engineering, is a hertuzumab-based HER2 ADC that consists of a cleavable mc-vc-PABC linker and MMAE payload (DAR = 4). Phase I trials of RC48 in HER2-positive advanced solid cancers demonstrated reasonable efficacy (ORR = 21% and disease control rate (DCR) = 49.1%) with a well-tolerated safety profile (NCT02881190). Most common ≥ grade 3 TRAEs included neutropenia, leukopenia, and hypoesthesia [103]. There is an ongoing phase III trial assessing RC48 efficacy in advanced GC (2.5 mg/kg Q2W) versus physician’s choice (NCT04714190) as well as numerous other trials in HER2-expressing cancers (NCT05996952, NCT05955209, NCT05980481, NCT0513715, NCT04329429). RC48 is also being examined in combination with immune therapies including PD-1 mAbs as well as the PD1xCTLA4 bsAb, cadonilimab (NCT06178601, NCT05912816, NCT05979740, NCT05726175).

#### 2.2.4. MRG002

MRG002, developed by Shanghai Miracogen Inc., consists of the parent mAb MAB802 and MMAE payload conjugated to interchain cysteines via a protease-cleavable vc linker (DAR = 3.8). MAB802 shares the same amino acid sequence as trastuzumab but is hyper-fucosylated to dampen ADCC activity and reduce the potential incidence of immune-related AEs. Importantly, preclinical evaluation of MRG002 in GC and BC PDX models demonstrated equivalent or superior efficacy over T-DM1 [104]. GLP toxicity studies in nonhuman primates determined the HNSTD as 6 mg/kg (Q3Wx4), and the major toxicities were reversible hematotoxicity and myelotoxicity. MRG002 is now undergoing several clinical trials (phases I through III) in HER2-positive and HER2-low cancers (NCT04941339, NCT04839510, NCT04742153).

#### 2.2.5. DP303c

Similar to MRG002 and RC48, DP303c, developed by CPSC ZhongQi Pharmaceutical Technology Co., Ltd., is a HER2-targeting, MMAE-conjugated ADC. DP303c incorporates a cleavable NH_2_-PEG3-vc linker via site-specific, microbial transglutaminase-mediated conjugation at Q295 in the mAb Fc region (DAR = 2). Compared with T-DM1, DP303c demonstrated equivalent or superior antitumor efficacy in GC and BC cell line xenografts (0.3–15 mg/kg similar to MRG002). This may be attributed in part to the fact that MMAE can exert a bystander effect, whereas DM1 cannot. Nonhuman primate studies found the HNSTD of DP303c to be 20 mg/kg (Q3Wx5), and major TRAEs were minimal to moderate lymphocyte depletion and lung inflammation, both of which were reversible [105]. DP303c has now progressed to phase III trials where it is being evaluated against T-DM1 in HER2-positive BC (NCT06313086).

#### 2.2.6. FS-1502

FS-1502 is a HER2-targeting ADC developed by Iksuda Biotherapeutics consisting of a cleavable β-glucuronide linker and MMAF payload. FS-1502 demonstrated reasonable antitumor activity in phase I clinical trials in HER2-expressing advanced solid tumors with an ORR of 66.7% (4/6) in patients receiving a dose greater than 1 mg/kg, and the recommended phase II dose (RP2D) was 2.3 mg/kg Q3W (NCT03944499). Grade 3 TRAEs were observed in 24/67 of patients, the most common being hypokalemia and decreased platelet counts [106]. FS-1502 is now being evaluated against T-DM1 (3.6 mg/kg) in a phase III trial in locally advanced or mBC (NCT05755048).

#### 2.2.7. Trastuzumab Duocarmazine (SYD985)

Trastuzumab duocarmazine, or SYD985, developed by Byondis, is composed of a trastuzumab backbone conjugated to the duocarmycin prodrug *seco*-duocarmycin-hydroxybenzamide-azaindole (*seco*-DUBA) via a cleavable vc linker (DAR = 2.8). Preclinical evaluation of SYD985 showed broader efficacy versus T-DM1 in BC PDX models, particularly in HER2-low models. A phase I dose-escalation study for SYD985 (NCT02277717) in several HER2-expressing cancers determined the RP2D at 1.2 mg/kg and the most common TRAEs included fatigue and dry eye with reasonable anti-tumor activity across HER2-positive and HER2-low tumors. SYD985 was assessed in the phase III TULIP trial versus physician’s choice in pretreated HER2-positive mBC and significantly improved PFS (7.0 months vs. 4.9 months) (NCT03262935) [107]. Presently, the FDA has issued a complete response letter for the Biological License Application of SYD985, suspending the approval decision for SYD985 in HER2-positive unresectable locally advanced or mBC [108].

#### 2.2.8. Emergent Clinical-Stage HER2-Targeting ADCs

In addition to the previously discussed phase III/approved HER2-targeted ADCs, there are several more clinical assets that have shown promising preliminary safety and efficacy. A166 and ZV0203, for example, are two HER2-targeted ADCs conjugated to a duostatin-5 (MMAF derivative) payload that have demonstrated efficacy in patients heavily pretreated with other HER2-targeted therapies, including T-DM1 and T-DXd, in phase I/II trials (NCT03602079; NCT05346328; NCT05423977) [109,110,111,112,113]. Similarly, ARX788, which consists of an MMAF payload conjugated via unnatural *para*-acetylphenylalanine amino acid conjugation, is being examined in the phase II ACE-Breast-03 clinical trial for HER2-positive mBC patients previously treated with T-DXd (NCT04829604) [114,115,116].

### 2.3. HER3-Targeting ADCs

In addition to EGFR and HER2, HER3-targeting ADCs have also been evaluated in the clinical setting (Table 1). HER3 is frequently upregulated in multiple cancer subtypes and correlates with poor disease prognosis [174,175,176]. Despite favorable safety profiles, the clinical progress of HER3-targeted mAbs has been stalled by a lack of objective responses [177,178,179,180]. Yet, HER3-targeted ADCs may be a more favorable option because of increased antitumor effects conferred by payload-mediated cytotoxicity.

#### 2.3.1. Patritumab Deruxtecan (U3-1402, HER3-DXd)

Patritumab deruxtecan, or U3-1402/HER3-DXd, developed by Daichii Sankyo in collaboration with Amgen, comprises the parent mAb patritumab conjugated to the cleavable GGFG tetrapeptide linker and topoisomerase I inhibitor DXd used in T-DXd [117]. The parent patritumab mAb competitively inhibits NRG1/2 binding to the HER3 ECD [118] and demonstrated reasonable safety and antitumor efficacy in phase I/II trials, although phase III trials were terminated because of a failure to meet pre-defined efficacy criteria, rationalizing the development of a patritumab-based ADC, HER3-DXd (NCT02134015; NCT01957280; NCT01211483) [118,119,177]. Preclinically, HER3-DXd exhibited durable responses and favorable toxicity profiles in lung, colorectal, breast, gastric, and melanoma cell line xenografts and PDX models (3–10 mg/kg) with negligible activity in HER3-low models [117,120,121]. Encouraging preclinical results prompted the phase I/II trial investigating HER3-DXd efficacy in heavily pretreated HER3-expressing mBC patients (NCT02980341) [122]. At lower doses (4.8 mg/kg), HER3-DXd exhibited a tolerable toxicity profile, with most common ≥ grade 3 TRAEs including decreased neutrophil, white blood cell, or platelet counts and anemia. Furthermore, HER-DXd achieved a median OS of 14.6 months in HR+/HER2− and triple-negative BC (TNBC) subtypes and 19.5% in HER2-positive BCs [123]. The ICARUS-BREAST01 phase II trial (NCT04965766), which evaluated HER3-DXd in heavily pretreated, HER3-unselected mBC patients, observed a DCR of 82.1% (40/56) [124]. The SOLTI TOT-HER3 (NCT0461528) study evaluated HER3-DXd in treatment-naïve patients with HR+/HER2− BC and reported meaningful clinical response following a single HER3-DXd dose (6.4 mg/kg) [125,126].

In addition to BC, clinical trials for HER3-DXd were also initiated in NSCLC. In a phase I dose escalation trial for EGFR TKI-pretreated metastatic NSCLC, HER3-Dxd (5.6 mg/kg Q3W) was well tolerated with a 39% ORR and median PFS of 8.2% (NCT03260491) [127]. In the dose expansion arm, HER3-Dxd (5.6 mg/kg) promoted a clinically meaningful response irrespective of RAS/PIK3CA/HER2 mutations, EGFR mutation, or c-MET amplification [128]. In the recently concluded phase II HERTHENA-Lung01 (NCT04619004) study, HER3-DXd dramatically improved ORR in EGFR-mutant NSCLC patients, including activity against brain metastases [129]. These data support the ongoing phase III HERTHENA-Lung02 (NCT05338970) trial evaluating HER3-Dxd versus standard-of-care chemotherapy in EGFR-mutated progressed NSCLC [130]. Yet, it is worth noting that ~60% of EGFR-mutated NSCLC patients do not respond to HER3-DXd because of low expression of HER3 [131]. Furthermore, HER3-DXd has been shown to be ineffective in other cancer subtypes, as indicated by the termination of a phase II trial examining HER3-DXd in advanced or metastatic CRC (NCT04479436).

#### 2.3.2. AMT-562

AMT-562, developed by Multitude Therapeutics, was generated by conjugating HER3-targeting mAb Ab562 to exatecan via a self-immolating PABC spacer [132]. Ab562 has moderate HER3 affinity, minimizing the potential for on-target toxicity and improving tumor penetration. AMT-562 also improved TGI in CRC cell line xenografts compared with HER3-DXd and Ab562-GGFG-DXd, demonstrating improved tumor-killing capacity conferred by Ab562 over patritumab [132]. Consistently, AMT-562 (5–10 mg/kg) demonstrated improved antitumor activity over HER3-DXd in several HER3-low xenograft models (5–10 mg/kg) and durable responses in HER3-Dxd insensitive CRC PDXs. AMT-562 also strongly synergized with VEGF- or EGFR-targeting mAbs bevacizumab and cetuximab, respectively, in CRC PDXs and with osimertinib or KRAS G12C inhibitor sotorasib in NSCLC PDX models [132]. Safety studies of AMT-562 in nonhuman primates determined the HNSTD as 30 mg/kg, with primarily gastrointestinal-related toxicities that were reversible following 1-week treatment withdrawal [132]. Taken together, these preclinical findings suggest that AMT-562 may be a clinically viable option for HER3-low patients and cancers including CRC. AMT-562 is now under evaluation in phase I clinical trials (NCT06199908).

#### 2.3.3. DB-1310

DB-1310, by Duality Biologics, is a HER3-targeting ADC produced by conjugating GenScript’s humanized 3F8 antibody via a cleavable maleimide tetrapeptide-based linker to the proprietary topoisomerase I inhibitor P1021 (DAR = 8) [133]. DB-1310 binds a distinct epitope from that of HER3-DXd with improved internalization and lysosomal trafficking [133]. Consistently, DB-1310 demonstrated increased TGI compared with HER3-DXd in lung, prostate, breast, and colon cancer cell line xenografts and PDXs (1–10 mg/kg) [133]. Similar to AMT-562, DB-1310 was also shown to synergize with osimertinib in EGFR-mutant NSCLC cell line xenografts and dramatically suppressed tumor growth in an osimertinib-resistant NSCLC PDX model [133]. Thus, DB-1310 may represent a viable clinical therapeutic option either in combination with or following patient progression on osimertinib. DB-1310 was well-tolerated in nonhuman primates (Q3Wx3) at an HNSTD of 45 mg/kg [133]. Lung and thymic toxicities were the primary observed TRAEs. Based on preclinical evaluation, a phase I/IIa trial of DB-1310 is underway in advanced/metastatic solid tumors (NCT05785741).

#### 2.3.4. YL202/BNT326

YL202/BNT326 is a HER3-targeting ADC developed by MediLink Therapeutics consisting of a cleavable tripeptide linker and a novel topoisomerase I inhibitor, YL0010014 (DAR = 8) [134]. Uniquely, YL-202 incorporates a TME-cleavable linker utilizing MediLink’s TMALIN platform. Pre-clinical evaluation of YL202 demonstrated dose-dependent antitumor activity in cancer cell line xenografts and PDXs of several cancer types and revealed a tolerable safety profile in nonhuman primates [135]. YL202 was subsequently evaluated in a phase I trial (NCT05653752) for NSCLC and HR+/HER2− mBC. Commonly observed TRAEs included anemia and hematopoietic toxicities associated with topoisomerase I inhibition. YL202 is now being evaluated in two separate phase II trials for several tumor types (NCT06107686; NCT06439771).

## 3. Bispecific ADCs Targeting the HER Family

While single-target ADCs have experienced great clinical success, their efficacy is limited by two main factors including drug resistance and TRAEs. ADC drug resistance is mediated through several mechanisms including upregulation of drug efflux pumps, antigen downregulation, and alterations in ADC processing [181]. For example, HER2 ADCs are often routed for membrane recycling rather than lysosomal degradation, resulting in premature release into the extracellular space [66,182,183]. Example resistance mechanisms are depicted in Figure 3. TRAEs are largely, though not exclusively, driven by ADC payloads, such as neutropenia for topoisomerase I inhibitors and hepatotoxicity and thrombocytopenia for maytansine derivatives [65]. One emerging strategy to overcome drug resistance and reduce TRAEs is the development of bsADCs that simultaneously target two surface antigens. Notably, while all bsADCs target two cognate antigens or antigen epitopes, their formats are incredibly diverse. Some commonly employed bsADC formats are depicted in Figure 4. Compared with monospecific ADCs, bsADCs may increase efficacy via enhanced internalization and lysosomal trafficking, more effective targeting of tumor heterogeneity, and direct targeting of drug resistance-mediating receptors [184]. Clinical-stage and select emergent pre-clinical bsADCs consisting of at least one EGFR/HER2/HER3 targeting arm are reviewed below and summarized in Table 2.

### 3.1. EGFR-Targeting bsADCs

#### 3.1.1. BL-B01D1 (EGFR x HER3)

BL-B01D1, developed by Systimmune, Inc. and Sichuan Baili Pharmaceutical Co., Ltd., is an EGFR x HER3 bsADC consisting of a cathepsin-cleavable tetrapeptide linker and a topoisomerase I-inhibiting camptothecin derivative payload, ED-04 (DAR = 8) [136]. BL-B01D1 consists of a biparatopic anti-EGFR Fab arm fused to two anti-HER3 single-chain fragment variables (scFv) by glycine–serine linkage, generating a tetravalent bsAb backbone [62]. Preclinical evaluation in lung, colorectal, and pancreatic cancer cell line xenografts demonstrated that BL-B01D1 improved cytotoxicity over EGFR or HER3 monospecific ADCs. Phase I trials for BL-B01D1 in locally advanced or metastatic solid tumors (NCT05194982) determined an RP2D of 2.5 mg/kg (D1D8Q3W). Commonly observed ≥ grade 3 TRAEs were predominantly hematopoietic and gastrointestinal as are commonly implicated with other topoisomerase inhibitor ADCs, such as SN38-loaded TROP2 ADC and sacituzumab govitecan [137,185]. Furthermore, 3% of patients (5/195) experienced serious AEs resulting in treatment discontinuation, and only one patient experienced ILD. Importantly, BL-B01D1 demonstrated promising antitumor activity with a DCR of 89%, ORR of 34%, and median PFS of 5.7 months [138]. BL-B01D1 is now being tested in phase I-III trials for different tumor indications (NCT05983432; NCT05880706; NCT06118333; etc.). Several trials are also examining the efficacy of BL-B01D1 in combination with a PD-1xCTLA4 bsAb (SI-B003) (NCT05990803; NCT05924841; NCT06006169). Of note, BCG019 is another EGFR x HER3 bsADC currently in preclinical development by Biocytogen conjugated to either MMAE or the novel topoisomerase I inhibitor BCPT02, although its characterization thus far is limited [139].

#### 3.1.2. M1231 (EGFR x MUC1)

M1231, a joint venture between Merck and Sutro Biopharma, is a bsADC targeting EGFR and mucin 1 (MUC1), which are highly co-expressed in cancers such as NSCLC, esophageal squamous cell carcinoma, HNSCC, TNBC, and ovarian cancer [140]. M1231 consists of an anti-MUC1 scFv and anti-EGFR Fab domain engineered by a SEED (strand-exchange engineered domain) antibody scaffold based on complementary IgG/IgA domains in the antibody CH3 Fc regions that promote heterodimerization [186]. M1231 consists of a cleavable vc-PABA linker conjugated via site-specific unnatural amino acids to SC209, a hemiasterlin derivative that disrupts microtubule dynamics [62]. M1231 improved internalization, lysosomal trafficking, and antitumor efficacy compared with monospecific EGFR or MUC1 bivalent antibodies, and pharmacokinetic (PK) studies in nonhuman primates determined an efficacious dose prediction range of 2.4–4.3 mg/kg Q3W [141]. M1231 has now completed phase I clinical trials (NCT04695847) for EGFR-expressing solid tumors, although results are not yet available nor are there ongoing phase II trials. Biocytogen has also reported the preclinical development of BSA01, another EGFR x MUC1 bsADC conjugated to MMAE [142,143].

#### 3.1.3. AZD9592 (EGFR x c-MET)

AZD9592, developed by AstraZeneca, is an EGFR x c-MET bsADC consisting of a proprietary topoisomerase I inhibitor, AZ14170132, conjugated to an EGFR x c-MET bsAb backbone via a cleavable peptide linker [144]. The EGFR x c-MET bsAb was constructed utilizing a DuetMab monovalent bispecific IgG platform that uses knobs-into-holes technology to promote heavy chain dimerization and, furthermore, reduces heavy and light chain mispairing via the introduction of an engineered disulfide bond in one of the CH1-CL interfaces. Importantly, the bsAb backbone demonstrates a higher binding affinity for c-MET so as to reduce EGFR-mediated toxicity [144]. Intriguingly, AZD9592 in vitro efficacy was most robust when both EGFR and c-MET were engaged, suggesting that bispecific targeting of EGFR and c-MET may increase tumor selectivity and reduce on-target toxicities [144]. AZD9592 showed efficacy in EGFR-mutated NSCLC and HNSCC PDX models (2 mg/kg) and in EGFR, ALK, and c-MET TKI-refractory PDX models [145,146]. Nonhuman primate safety studies revealed a well-tolerated safety profile consistent with other topoisomerase I inhibitors [144]. The ongoing EGRET trial (NCT05647122) is the first-in-human phase I study evaluating AZD9592 monotherapy and in combination with osimertinib in advanced/metastatic NSCLC and HNSCC [147].

#### 3.1.4. VBC101-F11 (EGFR x c-MET)

VBC101-F11 is another EGFR x c-MET bsADC developed by VelaVigo. Uniquely, VBC101-F11 is designed in a nanobody-based format (90kDa) that incorporates a low-affinity EGFR-targeting arm coupled to a high-affinity biparatopic c-MET arm and demonstrates superior cytotoxicity over the monospecific c-MET ADC ABBV-399 currently in phase III trials [148]. The nanobody format of VBC101-F11 also confers increased tumor penetration and accumulation compared with ABBV-399 and AZD9592 [148]. MMAE-conjugated VBC101-F11 displayed superior TGI over ABBV-399 in lung cancer cell line xenografts (3 mg/kg), and conjugation to a novel, undisclosed DNA replication inhibitor resulted in similar efficacy to AZD9592 in EGFR-low/c-MET-high and EGFR-/c-MET-moderate models [148]. The preclinical efficacy of VBC101-F11 supports its development in clinical trials, although phase I trials have not yet been initiated.

#### 3.1.5. DM001 (EGFR x TROP2)

DM001 is an EGFR x TROP2 bsADC developed by Biocytogen utilizing their RenLite antibody production platform that utilizes knobs-into-holes technology to facilitate heavy chain heterodimerization and incorporates a common light chain [149]. Notably, TROP2 ADCs such as FDA-approved sacituzumab govitecan (SN38 payload) and phase III datopotamab deruxtecan (DXd payload), have shown efficacy in HR+/HER2− mBC and TNBC or metastatic urothelial cancer, respectively (NCT05104866) [187]. EGFR and TROP2 are co-expressed in many solid tumors, and co-targeting these receptors may improve tumor selectivity compared with EGFR or TROP2 monospecific ADCs [149]. The DM001 EGFR x TROP2 bsAb was generated, which demonstrated comparable internalization to its parental EGFR or TROP2 monospecific parental mAbs. DM001 was conjugated to MMAE via a protease-cleavable linker to generate the DM001 bsADC, which showed more potent cell-killing capacity in EGFR/TROP2-positive cells than single-target positive cells and showed potent anti-tumor activity in lung and pancreatic cell line xenografts and PDXs [149]. Furthermore, the DM001 bsADC demonstrated superior efficacy over its parental monospecific EGFR- and TROP2 ADCs in lung and pancreatic cancer PDXs. At this point, Biocytogen has reported the submission of an Investigational New Drug application for DM001.

### 3.2. HER2-Targeting bsADCs

#### 3.2.1. Zanidatamab Zovodotin (ZW49; HER2 Biparatopic)

Zanidatamab zovodotin, or ZW49, developed by Zymeworks, is one of three biparatopic HER2-targeted ADCs to reach clinical trials. ZW49 consists of a zanidatamab (ZW25) mAb composed of an anti-HER2 ECDII Fab fragment and anti-HER2 ECDIV scFv fragment conjugated to a proprietary auristatin microtubule-inhibiting payload, ZD02044, via a cleavable dipeptide linker [150,151]. ZW25 induces receptor crosslinking and surface clustering, thereby enhancing internalization, lysosomal delivery, and receptor degradation [152]. Indeed, ZW49 internalized more rapidly than a monospecific trastuzumab-ZD02044 HER2 ADC. ZW49 also promoted tumor regression in HER2-high (3 mg/kg) and HER2-low (6 mg/kg) BC xenografts as well as GC PDXs (6 mg/kg) [151,153]. ZW49 was tolerable in a nonhuman primate toxicology study with no TRAEs observed up to the maximum dose tested (Q2Wx3; up to 12 mg/kg). A phase I dose escalation study for ZW49 for locally advanced or metastatic HER2-expressing cancers was initiated, with preliminary results determining a recommended dose as 2.5 mg/kg Q3W (ORR = 28%; DCR = 72%) (NCT03821233) [154]. Furthermore, 9% of patients had ≥ grade 3 TRAEs, including a grade 4 infusion reaction. Presently, plans for ZW49 phase II trials have been halted by Zymeworks pending further assessment of the clinical landscape [155].

#### 3.2.2. MEDI4276 (HER2 Biparatopic)

Like ZW49, MEDI4276 developed by AstraZeneca also targets HER2 ECDs II and IV. ZW49 is composed of the trastuzumab scFv and HER2-directed mAb 39S and is conjugated to a tubulysin variant, AZ13599185, via an mc protease-cleavable peptide-based linker attached utilizing site-specific (S239C and S442C) cysteine conjugation (DAR = 4) [156,157]. MEDI4276 also incorporates an L234F mutation in the Fc region to impair FcγR binding, potentially limiting FcγR-mediated toxicities like T-DXd- and T-DM1-associated ILD [98,188]. Importantly, MEDI4276 internalized faster with greater lysosomal trafficking and HER2 degradation versus monospecific HER2 mAbs. Furthermore, MEDI4276 promoted regression in T-DM1-resistant BC and GC xenografts and was tolerable in nonhuman primates with a safety profile similar to other microtubule-inhibiting agents (HNSTD = 1 mg/kg) [158]. These results motivated a phase I/II study of MEDI4276 in patients with HER2-expressing BCs or GCs (NCT02576548). However, ORR was only 9.4% and MEDI4276 was poorly tolerated, with 75% of patients experiencing one or more ≥ grade 3 TRAE at the maximum-tolerated dose of 0.75 mg/kg [158]. Five patients experienced TRAEs leading to discontinuation, including hepatotoxicity and peripheral neuropathy, which may potentially be attributed to on-target effects due to enhanced internalization and/or ECDII inhibition.

#### 3.2.3. JSKN003 (HER2 Biparatopic)

Developed by Alphamab Oncology, JSKN003 is the third HER2 biparatopic ADC in clinical trials and, like ZW49 and MEDI4276, targets HER2 ECDs II and IV. JSKN003 consists of a KN026 bsAb backbone conjugated to a topoisomerase I inhibitor via a dibenzocyclooctyne tetrapeptide linker by a proprietary glycan-specific conjugation strategy (DAR = 4) [159,160]. Notably, JSKN003 demonstrated more rapid internalization than T-DXd in GC and pancreatic cancer cell lines and promoted TGI in HER2-positive GC and BC cell line xenografts [159]. A first-in-human phase I study for JSKN003 (NCT05494918) was conducted in HER2-expressing patients, and interim analyses showed a well-tolerated safety profile with promising antitumor activity, particularly in HER2-amplified tumors (ORR = 71.4% for HER2 IHC 3+ patients; ORR = 55.6% and 37.5% for IHC 1+ and 2+, respectively). The maximum tolerated dose has not yet been reached after dosing up to 8.4 mg/kg [161]. JSKN003 is now under evaluation in phase I/II (NCT06226766; NCT05744427) tolerability and dose-escalation studies as well as phase III trials (NCT06079983) for HER2-low mBC.

#### 3.2.4. 23V-MMAE (HER2 x HER3)

To date, one preclinical HER2 x HER3 bsADC, 23V-MMAE developed by researchers at Shanghai Jiao Tong University, has been reported. 23V-MMAE utilizes bispecific antibody by protein trans-splicing (BAPTS) to join the anti-HER2 antibody sequence from trastuzumab and the anti-HER3 antibody sequence from DL11, an EGFR x HER3 bsAb [162]. The BAPTS platform joins two separately generated antibody fragments via autocatalytic protein trans-splicing of split inteins at the antibody hinge region and eliminates heavy or light chain mispairing [189]. Following protein trans-splicing, there is the insertion of a five-amino acid residue “CFNAS” sequence in the hinge region containing a cysteine with high oxidation efficacy, and V205C mutations were introduced into each antibody light chain for site-specific cysteine conjugation to an mc-vc-PAB-MMAE linker–payload, thus generating the 23V-MMAE bsADC (DAR = 2.89). The binding affinity of 23V-MMAE for HER2 and HER3 was comparable to that of parental anti-HER2 2V-MMAE and anti-HER3 3V-MMAE monospecific ADCs in BC cells and exhibited slightly improved internalization. Importantly, 23V-MMAE promoted more substantial TGI in T-DM1-resistant cells compared with 2V-MMAE or 3V-MMAE monotherapy. In BC cell line xenografts, TGI and the survival of 23V-MMAE-treated mice were comparable to 2V-MMAE and 3V-MMAE combination therapy (3 mg/kg) and 2V-MMAE monotherapy (10 mg/kg), although 3V-MMAE monotherapy had no effect. To date, 23V-MMAE is still in the preclinical stages, though this work provides proof of concept that HER2 x HER3 bsADCs may be more effective than monospecific ADCs.

#### 3.2.5. YH012 (HER2 x TROP2)

Additional HER2 bsADCs are being developed to enhance tumor specificity by co-targeting other non-RTK tumor-associated antigens such as TROP2, which is co-expressed with HER2 in various cancers [163]. YH012 is a HER2 x TROP2 developed by Biocytogen comprising an MMAE payload and protease-cleavable vc linker (DAR = 4). Importantly, the YH012 bsAb backbone demonstrated improved binding avidity and internalization compared with parental monospecific HER2 and TROP2 mAbs [163]. YH012 was also shown to be more selective for HER2+/TROP2+ cells rather than single target-positive cells, thereby enhancing tumor specificity and potentially reducing on-target toxicity compared with monospecific HER2 or TROP2 ADCs. Furthermore, YH012 induced potent TGI in NSCLC and GC cell line xenografts and HER2-low GC and CRC PDXs [163,164]. Nonhuman primate safety studies are still needed to determine the safety profile of YH012.

#### 3.2.6. BIO-201 (HER2 x TROP2)

BIO-201, a HER2 x TROP2 bsADC developed by BiOneCure Therapeutics, is conjugated to an undisclosed topoisomerase I inhibitor via a cleavable linker [165]. Preclinical characterization indicated that BIO-201 showed similar binding and internalization compared to parental HER2/TROP2 mAbs and promoted TGI in HER2- or TROP2-positive xenograft models similar to T-Dxd [165]. Notably, the efficacy of BIO-201 in HER2- or TROP2-positive xenografts may indicate broader clinical utility over traditional monospecific ADCs targeting HER2 or TROP2. Presently, BIO-201 is still in the preclinical phases of development.

### 3.3. HER3-Targeting bsADCs

#### 3.3.1. BCG022 (HER3 x c-MET)

BCG022, developed by Biocytogen, targets HER3 and c-MET, which are over-expressed in multiple solid tumors and mediate resistance to EGFR TKIs [180,190,191,192]. Similar to DM001, the BCG022 bsAb directed against HER3 and c-MET was generated using the RenLite production platform and showed superior internalization over HER3 or the c-MET parental mAbs. BCG022 was conjugated to either vc-MMAE or a BLD1102 linker-payload, which consists of a novel topoisomerase I inhibitor, BCPT02, and a highly hydrophilic protease-cleavable linker [166]. Both bsADCs induced robust tumor regression in NSCLC, GC, and pancreatic cancer xenografts, demonstrating potential clinical utility for BCG022-based bsADCs, although additional safety studies are warranted [166].

#### 3.3.2. DM002 (HER3 x MUC1)

DM002 developed by Biocytogen is a preclinical bsADC that simultaneously targets HER3 and MUC1, which are co-expressed in lung, gastric, breast, and pancreatic cancers. As with other Biocytogen-developed bsADCs (i.e., DM001 and BCG022), the DM002 bsAb was generated utilizing the RenLite production platform utilizing knobs-into-holes technology and conjugated to MMAE via a protease-cleavable linker [167]. Importantly, DM002 recognizes the juxtamembrane domain of MUC1 as high levels of MUC1 may induce auto-proteolysis and render MUC1 N-terminal-targeting antibodies ineffective. Preliminary characterization showed the DM002 bsADC induced marked antitumor effects in lung cancer, BC, GC, and pancreatic cancer cell line xenografts and PDXs [167]. Yet, as with the aforementioned Biocytogen bsADCs, DM002 is still in preclinical stages and warrants additional safety studies prior to entering clinical trials.

## 4. Emergent Trends

While ADCs have emerged as a prominent force in the field of anti-cancer therapeutics, the efficacy of FDA-approved ADCs is limited by TRAEs, underscoring the importance of novel strategies to improve tolerability and expand their therapeutic index. As previously outlined, bsADCs are one strategy that may hold significant potential to build on the success of monospecific ADCs. ADC combination therapies with agents such as TKIs, mAbs, chemotherapy, and immune checkpoint therapies are another potential strategy to improve ADC efficacy, as previously described. Other novel approaches toward advancing ADC development include novel payloads and ligand-targeting ADCs, which we briefly detail below as they pertain to HER family-targeted ADCs. These emergent trends are summarized in Figure 5.

### 4.1. ADCs Incorporating Novel Payloads

The majority of approved ADC payloads are either microtubule-inhibiting agents, DNA-damaging agents, or topoisomerase I inhibitors. However, the realm of alternative ADC payloads is rapidly expanding to include immune agonists, proteolysis-targeting activating chimeras (PROTACs), and BCL-XL and CDK inhibitors. Detailed below are several clinical and pre-clinical HER-targeting ADCs incorporating such novel payloads including toll-like receptor agonists and bromodomain-containing protein 4 (BRD4) PROTACs, among others.

#### 4.1.1. Immune-Stimulating Antibody Conjugates (ISACs)

HER2-targeting ADCs SBT6050, NJH395, and BDC-1001 are loaded with toll-like receptor 7/8 (TLR7/8) agonists that induce immune activation, also referred to as immune-stimulating antibody conjugates (ISACs). SBT6050 underwent phase I trials as a monotherapy and in combination with PD-1 mAbs pembrolizumab or cemiplimab (NCT04460456) and demonstrated a tolerable safety profile at 0.6 mg/kg Q2W according to interim analyses [193]. However, it appears that since these studies, SBT6050 development has been halted based on limited efficacy and cytokine-related TRAEs [194]. Similarly, NJH395 demonstrated severe cytokine-associated toxicities in phase I trials (NCT03696771) [195]. BDC-1001, consisting of a trastuzumab biosimilar backbone conjugated to a TLR7/8 agonist via a non-cleavable linker, is in phase I/II trials as monotherapy and in combination with nivolumab (NCT04278144) [196]. When dosed Q2W, BDC-1001 demonstrated a reasonable safety profile and efficacy up to 20 mg/kg, supporting further phase II expansion [197]. BDC-1001 recently began a separate phase II clinical trial in combination with pertuzumab (NCT05954143). Further studies are necessary to clarify the efficacy and safety of ISACs in the clinical setting across cancers.

#### 4.1.2. Degrader-Antibody Conjugates (DACs)

PROTACs are heterobifunctional molecules consisting of an E3 ligase-targeting moiety and a ligand for a protein of interest tethered together via a chemical linker often based on simple alkyl or PEG chains [198]. Functionally, PROTACs bring a protein of interest into proximity with the E3 ligase, promoting protein ubiquitination and proteasomal degradation. Much like ADCs, PROTAC linker design (i.e., length and composition) is a critical determinant of PROTAC efficacy, influencing binding cooperativity and protein isoform specificity [199]. Yet, PROTACs face challenges including cell permeability and lack of tissue specificity [200]. The conjugation of PROTACs to mAbs targeting tumor-enriched antigens, termed degrader–antibody conjugates, or DACs, may then increase the internalization and specificity of PROTACs. The potential of HER-targeting DACs was demonstrated with the preclinical development of a trastuzumab-based DAC conjugated to a BRD4 PROTAC (DAR = 4) that promoted BRD4 degradation exclusively in HER2-positive cells [201]. Furthermore, ORM-5029 is another DAC developed by Orum Therapeutics that consists of a pertuzumab backbone conjugated to a degrader of the translational termination factor GSTP1 (SMol006) via a cleavable vc-PABC linker [202]. Functionally, then, ORM-5029 exerts antitumor effects via GSPT1 degradation specifically in HER2-expressing cells. Preclinically, ORM-5029 demonstrated 10–1000 fold increased potency in HER2-expressing cell lines versus SMol006, T-DM1, and/or T-DXd monotherapies as well as antitumor efficacy in BC cell line xenografts superior to T-DM1 and comparable to T-DXd [202]. ORM-5029 has now begun phase I first-in-class, first-in-human clinical trials for HER2-expressing tumors (NCT05511844) and represents an exciting area for future collaboration between the ADC and PROTAC fields [203].

#### 4.1.3. ADCs Incorporating BCL-XL or CDK Inhibitors

Inhibitors of anti-apoptotic BLC-2 family members (i.e., BCL-2, BCL-XL) as well as CDK inhibitors have recently emerged as a potential ADC payload class. BCL-2 proteins play an important role in cancer by preventing apoptosis and promoting therapy resistance [204]. ABBV-637 is an EGFR-targeting ADC that consists of a BCL-XL inhibitor payload and has undergone phase I clinical trials in combination with osimertinib in EGFR-mutant NSCLC (NCT04721015). ABBV-637 was reported to show reasonable clinical activity (DCR = 73% or 65% for third- or second-line therapy, respectively) with a manageable safety profile, although phase II trials have yet to start [205]. Additionally, CDK inhibitors such as palbociclib, ribociclib, and ademaciclib have experienced great clinical success, though they are limited by normal tissue toxicity in animal models and patients [206,207,208]. To improve selectivity, an EGFR-targeting cetuximab-based ADC loaded with a selective CDK2/7/9 inhibitor, SNS-032, was recently developed and reported to inhibit tumor growth in EGFR-high TNBC xenografts [209]. Overall, novel payloads are an exciting avenue for ongoing and future ADC diversification.

### 4.2. Internalization-Enhancing bsADCs

For ADCs that rely on lysosomal degradation for payload release, internalization of the ADC–target complex is a crucial determinant of efficacy. Hence, there are ongoing efforts to increase the lysosomal trafficking of slow-internalizing receptors, such as members of the HER family, via the generation of bsADCs that target HER family members and other rapidly internalizing proteins. Though these types of bsADCs have yet to reach clinical evaluation, preclinical studies have shown potential translational promise.

Prolactin receptor (PRLR) has low expression in BC tumors and rapidly internalizes to the lysosome [210]. A HER2xPRLR bsADC with a DM1 payload (DAR = 3.3) and non-cleavable linker induced HER2 degradation and exhibited increased potency over parental ADC monotherapy and combination therapy in BC cells [210]. Similarly, the transmembrane amyloid precursor-like protein 2 (APLP2) is efficiently trafficked to lysosomes upon internalization [211]. A HER2xAPLP2 bsADC was developed via conjugation of trastuzumab to microtubule inhibitor dimethyl dolastatin 10 payload [211]. While the HER2xAPLP2 bsAb increased HER2 internalization and lysosomal trafficking versus trastuzumab, HER2xAPLP2 bsADC efficacy was inferior to a bivalent HER2 ADC in BC xenograft models, suggesting that HER2 bivalent engagement may be a more effective strategy for improved ADC efficacy.

Lysosome-associated membrane glycoprotein 3 (LAMP3) and sortilin-1 (SORT1) are protein transport regulators with roles in endocytosis. BsADCs targeting HER2 and LAMP3 or SORT1 were developed and conjugated to microtubule inhibitor duostatin-3 via a cleavable vc linker (DAR = 1) and DXd (DAR = 6.1), respectively [212,213]. Notably, the HER2xLAMP3 ADC utilized a low-affinity LAMP3 Fab to increase selectivity for cells expressing both targets as LAMP3 is ubiquitously expressed. Both bsAb backbones enhanced HER2 internalization in HER2-low cells, though not in HER2-high cells. Consistently, HER2xLAMP3 bsADC potency was similar to the bivalent HER2 ADC in HER2-high cells but showed enhanced TGI (8 mg/kg) compared with monospecific ADCs in HER2-moderate cells and xenografts [212]. Similarly, the HER2xSORT1 ADC promoted more marked TGI in HER2-low BC cell line xenografts (two weekly 10 mg/kg doses) compared with T-DXd and monospecific parental ADCs [213]. Thus, coupling HER family members with other rapidly internalizing proteins represents a promising avenue for enhanced ADC internalization and efficacy, particularly when HER family members are expressed at lower levels. At higher expression levels, though, bivalent HER engagement may be more effective.

### 4.3. HER Ligand-Targeted ADCs

Novel target antigens represent another opportunity to broaden ADC utility. Ligand-targeted ADCs are a unique subclass of ADCs directed against cell surface-tethered ligands enriched on tumor cells, such as inhibitory Notch ligand delta-like ligand 3 (DLL3) and Siglec-9 ligand galectin 3 binding protein (LGALS3BP) [214,215]. Within the HER family, some surface-tethered ligands are enriched on tumors and may make for suitable ADC targets [216,217,218,219]. For example, an AREG ADC, GMF-1A3-MMAE, is currently in preclinical investigation for endocrine-resistant BC [220]. GMF-1A3 recognizes the neo-epitope or transmembrane stalk of AREG that remains on the cell surface following its proteolytic cleavage. GMF-1A3-MMAE was successfully internalized to the lysosome and promoted tumor regression in BC cell line xenografts [220,221]. Similarly, our group recently reported EREG-targeting ADCs, including H231-EGC-qDuoDM-gluc, incorporating a stable tripeptide linker and self-immolative glucuronide-modified duocarmycin DM payload [222,223]. H231 recognizes the unprocessed proepiregulin and the mature, soluble EREG forms, thereby providing the additional benefit of EREG signaling neutralization mediated through EGFR and HER4. Furthermore, H231-EGC-qDuoDM-gluc promoted significant TGI in CRC cell line xenografts and PDX models. Importantly, targeting ligands that signal through multiple receptors may increase efficacy over traditional receptor-targeting ADCs.

## 5. Conclusions

It is evident that extensive work has been undertaken to develop safe and effective ADCs targeting members of the HER family. While HER2-directed T-DM1 and T-DXd are the only FDA-approved HER family-targeting ADCs, EGFR-targeting MRG003 and HER3-DXd in phase II and III trials, respectively, demonstrate potential clinical utility in targeting other HER family members. Importantly, while improving patient survival and disease control, HER-targeted ADCs have demonstrated mostly well-tolerated safety profiles, with most toxicity being driven by off-target, payload-mediated effects. MAb affinity maturation and Fc modifications, alterations in linker chemistry and stability, and payload diversification (i.e., tubulysin, eribulin, amanitin, and doxorubicin) have all been employed as attempts to improve the toxicities that are observed with ADC treatment and may further broaden their therapeutic indices [65,67]. Combination therapies with different immunotherapies, chemotherapies, TKIs, and mAbs may also represent a promising approach to improve the present success of ADCs. Furthermore, although not yet reported for ADCs, combination treatment regimens that target cancer-specific vulnerabilities (i.e., fasting to starve tumor cells of glucose and/or strategies to increase TME pH) may be worthwhile pursuits in improving ADC clinical efficacy. For example, bicarbonate improved response to several immunotherapies (anti-PD-1 and anti-CTLA4 mAbs and adoptive cell therapy) in melanoma xenograft models by increasing TME pH and counteracting immunosuppressive effects mediated by the acidic TME pH [224]. In addition to these strategies to improve current ADCs, there is a growing appreciation for determinants of efficacy that go beyond target expression, as exemplified by T-DXd efficacy in HER2-low and HER2-ultralow settings [173]. While the exact mechanisms are still unclear, these exciting developments seem to suggest that high antigen abundance is not required for ADC efficacy and may shed light on other important cell-killing mechanisms, such as the target-independent bystander effect. Lastly, while we have focused exclusively on HER-targeted ADCs in this review, it is worth noting that there are several other RTK-targeting ADCs in late-stage clinical trials, such as c-MET-targeting telisotuzumab vedotin and ROR1-targeting zilovertamab vedotin, both of which are currently being assessed in phase III trials (NCT04928846; NCT06093503; NCT05139017) [225,226]. Thus, as ADCs continue to expand rapidly in the anti-cancer therapeutic realm, the HER family and RTKs at large have vast potential to broaden ADC utility significantly and the portfolio of FDA-approved targets.

## Figures and Tables

**Figure 1 pharmaceutics-16-00890-f001:**
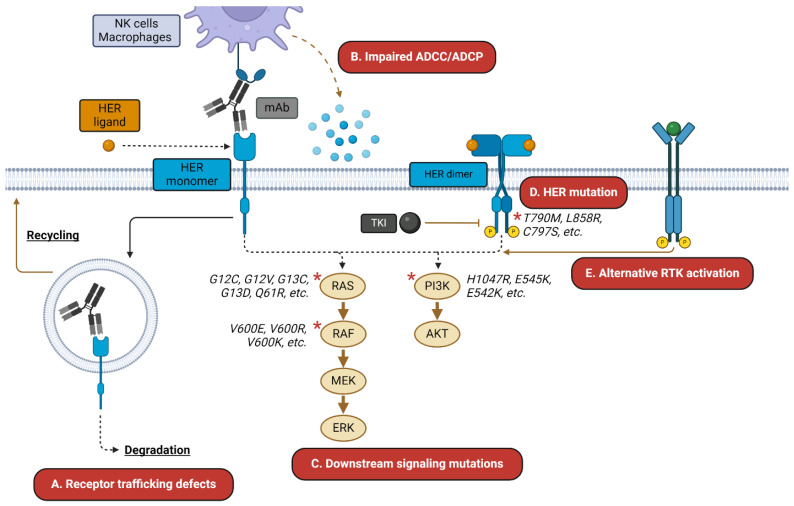
HER-targeted therapy resistance mechanisms. HER-targeted mAbs and TKIs are subject to a wide variety of resistance mechanisms. These may include (**A**) altered trafficking of the mAb–receptor complex away from degradation and toward recycling; (**B**) impaired ADCC/ADCP-mediated killing; (**C**) resistance mutations in downstream signaling mediators; (**D**) mutations in the receptor itself; and (**E**) activation of alternative receptors, such as c-MET. Asterisks indicate commonly mutated signaling mediators.

**Figure 2 pharmaceutics-16-00890-f002:**
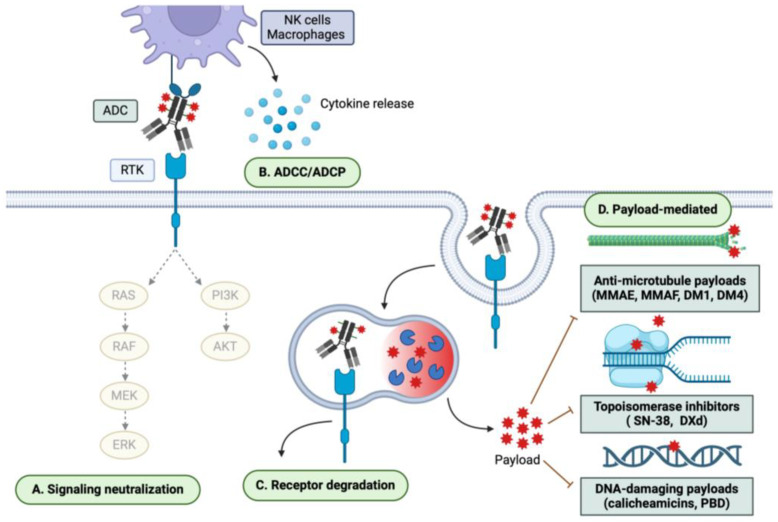
ADC mechanisms of cytotoxicity. ADCs exert both mAb- and payload-mediated effects. Therefore, the mechanisms of ADC cytotoxicity may include (**A**) neutralization of downstream signaling pathways; (**B**) ADCC/ADCP; (**C**) degradation of the target receptor; and (**D**) payload-mediated cytotoxicity, which may be mediated through anti-microtubule, topoisomerase-inhibiting, and DNA-damaging mechanisms, among others.

**Figure 3 pharmaceutics-16-00890-f003:**
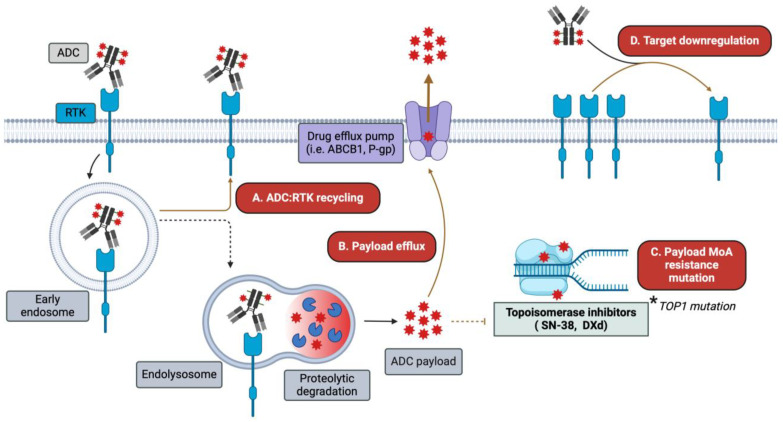
Mechanisms of ADC resistance. While ADCs circumvent some of the resistance mechanisms encountered by traditional HER-targeting mAbs and TKIs, alternative resistance mechanisms emerge. Among these are (**A**) increased ADC–receptor recycling/routing away from degradation pathways; (**B**) upregulation of drug efflux transporters (e.g., ABCB1 and P-gp); (**C**) mutations in the machinery involved in payload-mediated cytotoxicity, such as TOP1; and (**D**) target antigen downregulation following ADC treatment.

**Figure 4 pharmaceutics-16-00890-f004:**
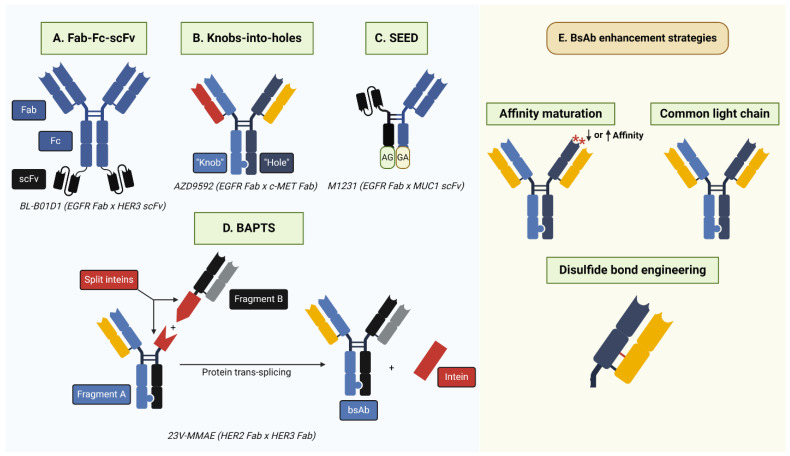
BsAb generation strategies. (**A**) Fab-Fc-scFv involves the fusion of Fab and scFv antibody fragments targeting separate antigens. Notably, the scFv region may be attached to regions other than those depicted above, such as the Fab N-terminus. (**B**) Knobs-into-holes require the incorporation of complementary “knob” (T366W) and “hole” (T366S; L368A; Y407V) mutations into the antibody CH3 region to facilitate heterodimerization. BsAbs may have common or different light chains. (**C**) Strand-exchange engineered domain (SEED) relies on the generation of CH3 domains with alternating IgG/IGA domains (“GA”/“AG”) that facilitate heterodimerization. SEED technology is also amenable to the incorporation of alternative antibody formats, such as scFv, as depicted above. (**D**) Bispecific Antibody by Protein Trans-Splicing (BAPTS) generates two antibody fragments fused to the split intein Npu DnaE. Following protein-splicing, the full-length intein is removed and the full-length bsAb is generated. (**E**) Common strategies to enhance bsAb efficacy, selectivity, and production include affinity maturation to alter antibody affinity for a target antigen and the incorporation of a common light chain or engineered cysteine/disulfide bond to eliminate heavy–light chain mispairing. Asterisks depict amino acid mutations in the complementarity-determining region.

**Figure 5 pharmaceutics-16-00890-f005:**
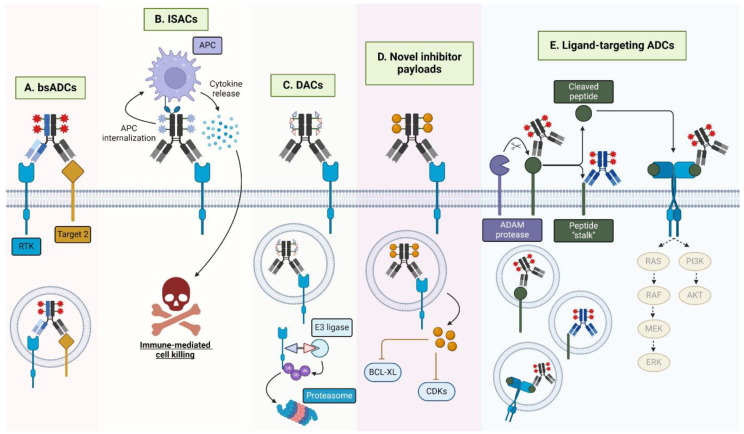
Emergent HER-targeting antibody-conjugate modalities. (**A**) BsADCs engage two tumor-associated antigens, such as two RTKs. The development of bsADCs that engage a rapidly internalizing receptor is another common motif for bsADCs. (**B**) ISACs, loaded with immune agonists, such as TLR7/8 agonists, engage the FcγR on antigen-presenting cells (APCs), mediating internalization, immune activation, and cytokine release, resulting in target cell death. (**C**) DACs are loaded with protein degraders such as PROTACs or molecular glues that, upon internalization, mediate ubiquitination and subsequent proteasomal degradation of the degrader-targeted protein (i.e., BRD4, GSPT1). (**D**) ADCs with novel inhibitors, such as BCL-XL and CDK inhibitors function by specifically targeting these cytotoxins to target-expressing cells. (**E**) Ligand-targeted ADCs may recognize full-length surface-bound ligands, solubilized ligands, or novel ligand epitopes remaining following ligand proteolytic cleavage. Furthermore, ligand-targeted ADCs may have the added benefit of signaling neutralization through the associated ligand’s receptor.

**Table 1 pharmaceutics-16-00890-t001:** HER-targeting monospecific ADCs.

**EGFR-Targeted ADCs**								
**Name**	**Developer**	**Payload**	**Linker**	**Cleavable?**	**Conjugation**	**DAR**	**Clinical Stage**	**Ref.**
ABT-414	Abbvie	MMAF	mc	N	Interchain cysteines	3.8	Completed Phase III	[77,78,79,80,81,82,83]
ABBV-221	Abbvie	MMAE	mc-vc-PABC	Y	Interchain cysteines	3	Discontinued	[84,85,86]
ABBV-321	Abbvie	PBD	mc-va	Y	Site-specific (S238C)	2	Discontinued	[87]
MRG003	Shanghai Miracogen	MMAE	mc-vc-PABC	Y	UD	UD	Phase II	[88]
HLX42	Henlius	Topo I inhibitor	UD	Y	UD	8	Phase I	[89]
**HER2-Targeted ADCs**								
**Name**	**Developer**	**Payload**	**Linker**	**Cleavable?**	**Conjugation**	**DAR**	**Clinical Stage**	**Ref.**
T-DM1	Genentech ImmunoGen	DM1	Thioether	N	Lysine	3.5	Approved	[90,91,92,93,94]
T-DXd	Daichii Sankyo AstraZeneca	DXd	GGFG	Y	Interchain cysteines	8	Approved	[95,96,97,98,99,100,101,102]
RC48	Remegen Biosciences Yantai Rongchang Biological Engineering	MMAE	mc-vc-PABC	Y	Interchain cysteines	4	Phase III	[103]
MRG002	Shanghai Miracogen	MMAE	mc-vc-PABC	Y	Interchain cysteines	3.8	Phase III	[104]
DP303c	CSPC ZhongQi Pharmaceutical Technology	MMAE	NH2-PEG3-vc	Y	mTG site-specific	2	Phase III	[105]
FS-1502	Iksuda Biotherapeutics	MMAF	β-glucuronide	Y	Prenyl transferase site-specific	2	Phase III	[106]
SYD985	Byondis	*seco*-DUBA	vc	Y	Interchain cysteines	2.8	Completed Phase III	[107,108]
A166	Klus Pharma	Duostatin-5	vc	Y	Site-specific	2	Phase II	[109,110,111]
ZV0203	Hangzhou Adcoris Biopharma	Duostatin-5	vc	Y	UD	2	Phase I	[112,113]
ARX788	Zheijang Medicine Ambrx	MMAF	Hydroxylamine-PEG4	N	Unnatural amino acid	1.9	Phase II	[114,115,116]
**HER3-Targeted ADCs**								
**Name**	**Developer**	**Payload**	**Linker**	**Cleavable?**	**Conjugation**	**DAR**	**Clinical Stage**	**Ref.**
HER3-DXd	Daichii Sankyo Amgen	DXd	GGFG	Y	Interchain cysteines	8	Phase III	[117,118,119,120,121,122,123,124,125,126,127,128,129,130,131]
AMT-562	Multitude Therapeutics	Exatecan	mc-va-T800	Y	Site-specific	8	Phase I	[132]
DB-1310	Duality Biologics	P1021	Maleimide tetrapeptide	Y	UD	8	Phase I/IIa	[133]
YL202/BNT326	MediLink Therapeutics	YL0010014	Tripeptide	Y	UD	8	Phase II	[134,135]

UD, Undisclosed.

**Table 2 pharmaceutics-16-00890-t002:** HER-targeting bsADCs.

Name	Developer	Targets	Structure	Strategy	Payload	Linker	Cleavable?	Conjugation	DAR	Clinical Stage	Ref.
BL-B01D1	SystImmune Sichuan Baili Pharmaceutical	EGFR x HER3	EGFR Fab x HER3 scFv	Fab-scFv	ED-04	Tetrapeptide	Y	UD	8	Phase III	[136,137,138]
BCG019	Biocytogen	EGFR x HER3	EGFR Fab x HER3 Fab	Knobs-into-holes (RenLite)	MMAE BCPT02	vc UD	Y Y	UD	4 UD	Preclinical	[139]
M1231	Merck Sutro Biopharma	EGFR x MUC1	EGFR Fab x MUC1 scFv	SEED-antibody scaffold	SC209	vc-PABA	Y	Site-specific, unnatural amino acid	4	Phase II	[140,141]
BSA01	Biocytogen	EGFR x MUC1	EGFR Fab x MUC1 Fab	Knobs-into-holes (RenLite)	MMAE	vc	Y	UD	4	Preclinical	[142,143]
AZD9592	AstraZeneca	EGFR x c-MET	EGFR Fab x c-MET Fab	Knobs-into-holes (DuetMab)	AZ14170132	Peptide-based	Y	UD	UD	Phase I	[144,145,146,147]
VBC101-F11	VelaVigo	EGFR x c-MET	Nanobody-based	UD	MMAE	UD	UD	UD	4	Preclinical	[148]
DM001	Biocytogen	EGFR x TROP2	EGFR Fab x TROP2 Fab	Knobs-into-holes (RenLite)	MMAE	vc	Y	UD	4	IND submitted	[149]
ZW49	Zymeworks	HER2 biparatopic	HER2 ECDII Fab x HER2 ECDIV scFv	UD	ZD02044	Dipeptide	Y	Interchain cysteine	UD	Halted	[150,151,152,153,154,155]
MEDI4276	AstraZeneca	HER2 biparatopic	HER2 ECDII Fab x HER2 ECDIV scFv	Fab-scFv	AZ13599185	mc	Y	Site-specific cysteine	4	Phase I/II	[156,157,158]
JSKN-003	Alphamab Oncology	HER2 biparatopic	HER ECDII Fab x HER2 ECDIV Fab	UD	Topo I inhibitor	Tetrapeptide	Y	Site-specific glycan	4	Phase III	[159,160,161]
23V-MMAE	Shanghai Jiao Tong University Jecho Institute Co., Ltd.	HER2 x HER3	HER2 Fab x HER3 Fab	MMAE	mc-vc-PAB	Dipeptide	Y	Interchain cysteine	2.89	Preclinical	[162]
YH012	Biocytogen	HER2 x TROP2	HER2 Fab x TROP2 Fab	Knobs-into-holes (RenLite)	MMAE	vc	Y	UD	4	Preclinical	[163,164]
BIO-201	BiOneCure Therapeutics	HER2 x TROP2	UD	UD	Topo I inhibitor	UD	Y	UD	UD	Preclinical	[165]
BCG022	Biocytogen	HER3 x c-MET	HER3 Fab x c-MET Fab	Knobs-into-holes (RenLite)	MMAE BCPT02	vc UD	Y Y	UD	4 UD	Preclinical	[166]
DM002	Biocytogen	HER3 x MUC1	HER3 Fab x MUC1 Fab	Knobs-into-holes (RenLite)	MMAE	vc	Y	UD	4	Preclinical	[167]

UD, undisclosed.

## Data Availability

Not applicable.

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
