# Peer review of "The Evolving Paradigm of Antibody–Drug Conjugates Targeting the ErbB/HER Family of Receptor Tyrosine Kinases"

_pharmaceutics, 2024, doi:10.3390/pharmaceutics16070890_

Round 1

Reviewer 1 Report

Comments and Suggestions for Authors

The manuscript presents a timely well-organised review on ADCs targeting the ErbB/HER Family of Receptor Tyrosine Kinases, which is of interest to the readership of Pharmaceutics.

Some amendments are recommended before publication:

1. In l. 166, the term "explosion" is not appropriate.

2. The authors describe numerous examples that (pre)clinical results of drug candidates are often disappointing compared with encouraging preclinical results, in particular dur to various adverse effects, This is not a rare outcome, e.g. as frequently observed regarding various immunotherapies based on mAbs. In this respect, it is suitable to emphasize that the therapeutic success of the standard cancer therapies is insufficient as can be deduced from the increasing number of patients, etc.

Therefore, it will be of interest to briefly discuss (particularly in 4. Emerging trends and 6. Conclusions) combined therapies with simple but effective measures addressing metabolic peculiarities of tumor cells, e.g. fasting and strict avoidance of sugar uptake (i.e. cutting off tumor cells from glucose as an essential nutrient) or increasing the pH level in the TME (as tumors can only grow in an acidic environment). Beneficial effects in combination with chemotherapies or as exclusive therapy are well documented. If not yet reported, such therapeutic strategies can be concisely suggested.

Author Response

Comments 1: In l. 166, the term "explosion" is not appropriate. 

Response 1: Thank you for pointing this out. Lines 183 now reads as “The rapid growth of HER2-targeting ADCs …”

Comments 2: The authors describe numerous examples that (pre)clinical results of drug candidates are often disappointing compared with encouraging preclinical results, in particular due to various adverse effects. This is not a rare outcome, e.g. as frequently observed regarding various immunotherapies based on mAbs. In this respect, it is suitable to emphasize that the therapeutic success of the standard cancer therapies is insufficient as can be deduced from the increasing number of patients, etc.

Therefore, it will be of interest to briefly discuss (particularly in 4. Emerging trends and 6. Conclusions) combined therapies with simple but effective measures addressing metabolic peculiarities of tumor cells, e.g. fasting and strict avoidance of sugar uptake (i.e. cutting off tumor cells from glucose as an essential nutrient) or increasing the pH level in the TME (as tumors can only grow in an acidic environment). Beneficial effects in combination with chemotherapies or as exclusive therapy are well documented. If not yet reported, such therapeutic strategies can be concisely suggested.

Response 2: Thank you for the thoughtful suggestion. While these sorts of treatments have yet to be reported in regard to ADCs, we have added the following in the conclusions section, lines 832-879: “Furthermore, although not yet reported for ADCs, combination treatment regimens that target cancer-specific vulnerabilities (i.e. fasting to starve tumor cells of glucose and/or strategies to increase TME pH) may be worthwhile pursuits in improving ADC clinical efficacy. For example, bicarbonate improved response to several immunotherapies (anti-PD-1 and anti-CTLA4 mAbs and adoptive cell therapy) in melanoma xenograft models by increasing TME pH and counteracting immunosuppressive effects mediated by the acidic TME pH [228].”

Reviewer 2 Report

Comments and Suggestions for Authors

This review provides a comprehensive description of the panorama of mono- and bi-specific ADCs targeting receptor tyrosine kinases of the ErbB/HER family. The information reported for FDA-approved and clinical trial ADCs is up-to-date and accurate, which is not easy given this rapidly evolving topic. Overall, the review is well written, the references cited are appropriate and the figures are easy to understand. It is recommended that the manuscript be accepted for publication in the present form.

Author Response

Comments 1: This review provides a comprehensive description of the panorama of mono- and bi-specific ADCs targeting receptor tyrosine kinases of the ErbB/HER family. The information reported for FDA-approved and clinical trial ADCs is up-to-date and accurate, which is not easy given this rapidly evolving topic. Overall, the review is well written, the references cited are appropriate and the figures are easy to understand. It is recommended that the manuscript be accepted for publication in the present form.

Response 1: Thank you for your time taken to review this manuscript. Your comments are well-received and appreciated.

Reviewer 3 Report

Comments and Suggestions for Authors

The manuscript titled "The Evolving Paradigm of Antibody-Drug Conjugates Targeting the ErbB/HER 2 Family of Receptor Tyrosine Kinases" provides a comprehensive review of the current state of antibody-drug conjugates (ADCs) targeting the HER family of receptor tyrosine kinases. The detailed discussion on clinical-stage ADCs targeting EGFR, HER2, and HER3, as well as novel bispecific ADCs (bsADCs), is particularly valuable. Additionally, the manuscript covers emerging trends in ADC development, including novel payloads and HER ligand-targeted ADCs, which are crucial for advancing cancer therapeutics. However, there are several areas where the manuscript could be improved to provide a more thorough and up-to-date overview of this rapidly evolving field.

1. Discontinuation of Clinical Trials for Zanidatamab Zovodotin:

   - The current manuscript discusses the ongoing clinical trials for Zanidatamab Zovodotin (ZW49). However, it should be updated to reflect the latest information indicating that these trials have been discontinued. This is critical information for readers and is necessary to provide an accurate picture of the current state of ADC clinical development.

2. Detailed Targeting Strategies:

   - The manuscript would benefit from a more detailed explanation of the specific targeting strategies for each member of the HER family. 

3. Side Effects and Safety:

   - More detailed information on the side effects observed in clinical trials of ADCs and their management strategies should be provided. This would help to clarify the safety profile of ADCs and their potential for clinical application.

Additional Comments for Future Direction of ADCs:

4. Site-Specific Conjugation (SSC):

   - The manuscript lacks detailed discussion on the advantages of SSC techniques. SSC ensures that the cytotoxic drug is attached to the antibody at a specific site, enhancing the stability and efficacy of ADCs while minimizing off-target effects. Discussing chemical site-specific modification methods and tag-free enzymatic modification methods, which have been highlighted in recent reviews, would be beneficial. These methods have shown promise in increasing the therapeutic window of ADCs by improving their pharmacokinetic (PK) and pharmacodynamic (PD) properties.

5. Novel Payloads:

   - The manuscript should have comments on the potential of novel payloads beyond traditional cytotoxic agents. Recent advancements have introduced payloads that target different cellular mechanisms, such as RNA polymerase inhibitors and immunomodulatory agents. These new payloads offer opportunities to overcome resistance mechanisms and enhance the efficacy of ADCs in treating various cancers, including ovarian cancer. Referencing and citing comprehensive reviews on this topic would strengthen this discussion (https://www.tandfonline.com/doi/abs/10.1080/14712598.2023.2276873).

Author Response

Comments 1: The current manuscript discusses the ongoing clinical trials for Zanidatamab Zovodotin (ZW49). However, it should be updated to reflect the latest information indicating that these trials have been discontinued. This is critical information for readers and is necessary to provide an accurate picture of the current state of ADC clinical development.
Response 1: Thank you for pointing this out. The manuscript and Table 1 have now been updated to reflect the present status of ZW49 (lines 603-604).

Comments 2: The manuscript would benefit from a more detailed explanation of the specific targeting strategies for each member of the HER family. Response 2: Thank you for the suggestion. We have included additional details for approved HER family-targeting TKIs and mAbs including binding mechanism, binding epitopes and mAb structures (lines 91-93; 97-98; 111-118).

Comments 3: More detailed information on the side effects observed in clinical trials of ADCs and their management strategies should be provided. This would help to clarify the safety profile of ADCs and their potential for clinical application.

Response 3: Thank you for the thoughtful suggestion. We have added in additional information regarding commonly observed treatment-related adverse events (TRAEs) associated with approved payload classes as well as management strategies utilized to remedy these (lines 153-159).

Comments 4: The manuscript lacks detailed discussion on the advantages of SSC techniques. SSC ensures that the cytotoxic drug is attached to the antibody at a specific site, enhancing the stability and efficacy of ADCs while minimizing off-target effects. Discussing chemical site-specific modification methods and tag-free enzymatic modification methods, which have been highlighted in recent reviews, would be beneficial. These methods have shown promise in increasing the therapeutic window of ADCs by improving their pharmacokinetic (PK) and pharmacodynamic (PD) properties.

Response 4: Thank you for pointing this out – SSC is an important, recent development in the ADC field. We have added additional information about the benefits of SSC in relation to traditional stochastic conjugation methods (lines 159-164).

Comments 5: The manuscript should have comments on the potential of novel payloads beyond traditional cytotoxic agents. Recent advancements have introduced payloads that target different cellular mechanisms, such as RNA polymerase inhibitors and immunomodulatory agents. These new payloads offer opportunities to overcome resistance mechanisms and enhance the efficacy of ADCs in treating various cancers, including ovarian cancer. Referencing and citing comprehensive reviews on this topic would strengthen this discussion (https://www.tandfonline.com/doi/abs/10.1080/14712598.2023.2276873).

Response 5: Thank you for this suggestion. We have included these novel payloads in lines 150-153 and revisit their potential to build upon the present success of traditional payloads in the conclusion.

Round 2

Reviewer 3 Report

Comments and Suggestions for Authors

Accept in present form